# Abscisic Acid Inhibits Cortical Microtubules Reorganization and Enhances Ultraviolet-B Tolerance in *Arabidopsis thaliana*

**DOI:** 10.3390/genes14040892

**Published:** 2023-04-10

**Authors:** Lichun Shi, Kun Lin, Tongbing Su, Fumei Shi

**Affiliations:** 1School of Life Science, Liaocheng University, Liaocheng 252059, China; 2National Engineering Research Center for Vegetables, Beijing 100097, China; 3Beijing Key Laboratory of Vegetable Germplasms Improvement, Beijing 100097, China; 4Key Laboratory of Biology and Genetics Improvement of Horticultural Crops (North China), Beijing 100097, China; 5Beijing Vegetable Research Center (BVRC), Beijing Academy of Agriculture and Forestry Science (BAAFS), Beijing 100097, China

**Keywords:** ultraviolet-B, abscisic acid, cortical microtubules, root growth, *Arabidopsis thaliana*

## Abstract

Ultraviolet-B (UV-B) radiation is one of the important environmental factors limiting plant growth. Both abscisic acid (ABA) and microtubules have been previously reported to be involved in plant response to UV-B. However, whether there is a potential link between ABA and microtubules and the consequent signal transduction mechanism underlying plant response to UV-B radiation remains largely unclear. Here, by using *sad2-2* mutant plants (sensitive to ABA and drought) and exogenous application of ABA, we saw that ABA strengthens the adaptive response to UV-B stress in *Arabidopsis thaliana* (*A. thaliana*). The abnormal swelling root tips of ABA-deficient *aba3* mutants demonstrated that ABA deficiency aggravated the growth retardation imposed by UV-B radiation. In addition, the cortical microtubule arrays of the transition zones of the roots were examined in the *aba3* and *sad2-2* mutants with or without UV-B radiation. The observation revealed that UV-B remodels cortical microtubules, and high endogenous ABA can stabilize the microtubules and reduce their UV-B-induced reorganization. To further confirm the role of ABA on microtubule arrays, root growth and cortical microtubules were evaluated after exogenous ABA, taxol, and oryzalin feeding. The results suggested that ABA can promote root elongation by stabilizing the transverse cortical microtubules under UV-B stress conditions. We thus uncovered an important role of ABA, which bridges UV-B and plants’ adaptive response by remodeling the rearrangement of the cortical microtubules.

## 1. Introduction

Climate change and the frequent activities of humans have intensified the depletion of the ozone layer in the atmosphere, resulting in enhanced ultraviolet radiation reaching the Earth’s surface. Plant growth and development are inevitably affected by ultraviolet-B (UV-B) radiation (280–320 nm), which has prime damaging effects among ultraviolet radiations of 200–400 wavelengths and has been a growing concern for years. Enhanced UV-B has severely harmful effects on plant growth and development by prompting DNA damage, membrane changes, and protein crosslinking [1,2,3], causing changes in vital physiology processes, such as decreased photosynthetic rate [4] and plant morphology along with hormonal and signal shifts [5,6,7]. However, the underlying physiological and molecular mechanisms in plants remain unclear.

As a crucial organ for sessile plants, the root system response to UV-B radiation is reported widely. UV-B irradiation has been shown to retard the growth of barley roots and induce subapical root swelling, which leads to cortical cell expansion and subsequent swelling of the lower part of the elongation zone [8]. The *A. thaliana* mutants *rus1* and *rus2,* root UV-B sensitive (roots sensitive to UV-B), show stunted root growth under low UV-B radiation [9]. In particular, the photoreceptor UV RESISTANCE LOCUS 8 (UVR8) responds to low doses of UV-B radiation and is confirmed to be expressed throughout the plant, including in *A. thaliana* roots [10,11]. UVR8 acts as a UV-B photoreceptor that interacts with MYB73/MYB77 (MYB domain protein 73/77) to regulate auxin response and lateral root development [12]. Inhibition of primary root elongation by UV-B is independent of the UVR8 [13]. The mechanism of the root response to UV-B radiation is involved in early seedling morphogenesis and development, which is closely related to hormones, such as abscisic acid, nitric oxide, auxin, etc. [4,5,6].

Abscisic acid (ABA) is a significant stress hormone and regulates many plant growth processes [14,15,16,17]. ABA was considered a growth inhibitor, negatively regulating plant development, such as root growth [17]. On the other hand, higher ABA concentrations are critical for the adaptive response of maize to UV-B irradiation. This response occurs via hydrogen peroxide generation and nitric oxide production, which maintains cell homeostasis and attenuates UV-B-derived cell damage [5]. ABA-related mutants are consequently isolated and are becoming powerful tools for studying *A. thaliana’s* response to stress. For instance, the ABA-deficient mutant, *aba3*, is involved in cold and osmotic stress response [18] and contributes to oxidative stress tolerance [7]. Another mutant, *sad2*, sensitive to ABA and drought, shows enhanced tolerance to UV-B radiation [19,20]. We thus believe that ABA is needed in UV-B signaling transduction and is worth verifying in detail.

As a highly conserved and dynamic subcellular complex, the microtubule cytoskeleton is adaptively rearranged in response to almost all kinds of stimuli [21,22,23,24,25]. Plant growth and morphogenesis change due to disturbed microtubule organization in *A. thaliana* which suffered from UV-B radiation [6,26]. Microtubule reorganization and corresponding morphology caused by stress are connected with cellular signals, such as nitric oxide [6,24,27], brassinosteroid [28], and ABA [29,30]. It is essential to comprehensively investigate the effects of UV-B radiation on plant growth and the relations of hormone signals and microtubule cytoskeleton.

In summary, the detailed cytological processes and mechanisms underlying the plant response to UV-B radiation remain largely unclear. In the study, we focused on the joint effects of ABA, microtubule dynamics, and potential crosstalk on the root response to UV-B radiation. We uncovered an important role of ABA bridges concerning UV-B and plants’ adaptive response by remodeling the rearrangement of the cortical microtubules.

## 2. Materials and Methods

### 2.1. Plants and Growth Conditions

The *A*. *thaliana* Columbia (Col-0) ecotype was used as the wild type for all experiments in this study. The *Arabidopsis thaliana* mutant *aba3* (a mutant deficient in ABA synthesis) [18] and the mutant *sad2-2* (sensitive to ABA and drought and tolerant to UV-B) [19,20] were used. The seeds were sterilized and incubated at 4 °C in the dark for 2 days in a Murashige & Skoog (MS) basal medium containing 1.5% sucrose (*w*/*v*) and 0.7% agar (*w*/*v*) and were placed in the growth cabinet under a 16:8 h light/dark cycle at 22 °C for 5 days and then harvested for experiments.

### 2.2. UV-B and Pharmaceutical Treatments

Healthy seedlings were irradiated with 90 mJ cm^−2^ of UV-B for 30 min. A spectral irradiance of 90 mJ cm^−2^ was determined with an Ultraviolet Crosslinker (midrange, 302 nm; UVP Co., Upland, CA, USA). The 30 min irradiation duration was selected because it was the shortest UV-B exposure period out of the 0 to 60 min monitored periods, which affected significant phenotypes (Figure A1). After radiation, the seedlings were positioned upside down for phenotype analysis in six days, and their cortical microtubules were observed 1, 2, and 6 h post-radiation.

Six-day-old vertically grown seedlings were transferred to an MS medium supplemented with 0, 0.25 and 0.5 μmol L^−1^ ABA, 1 μmol L^−1^ paclitaxel (Taxol^®^, Sigma, St. Louis, MO, USA), and 0.3 μmol L^−1^ oryzalin (3, 5-dinitro-*N*4, *N*4-dipropyl sulfanilamide, Sigma, St. Louis, MO, USA). They were positioned upside down for growth and were subsequently used to observe cortical microtubules; they were analyzed at 24, 36, and 48 h post-transfer.

### 2.3. ABA Extraction and Quantification

Healthy seedlings that did or did not receive UV-B treatment were collected and ground in a cold extraction buffer containing 80% methanol and 1 mmol of L^−1^ butylated hydroxytoluene. After 4 h of shaking in the dark at 4 °C, the homogenates were centrifuged (8400× *g*, 20 min, 4 °C), and the pellets were re-extracted for 60 min in 1 mL of the same extraction solvent. The supernatants were transferred to a fresh glass tube and vacuum-dried using Power Dry LL3000 (Heto-Holten, Allerød, Denmark). Extracts were dissolved and diluted in PBS buffer containing 1 mL L^−1^ Tween-20 and 1 g L^−1^ gelatin. ABA contents were determined using a Plant Hormone ABA ELISA kit (R&D, Emeryville, CA, USA) according to the manufacturer’s instructions.

### 2.4. Cortical Microtubules Immunolabeling and Observations

Seedling roots exposed or not to the above-described treatment were fixed with 4% (*w*/*v*) paraformaldehyde to investigate the effects of UV-B radiation and ABA on the cortical microtubule array. A mouse monoclonal antibody against β-tubulin (Sigma, St. Louis, MO, USA) at a 1:300 dilution and an Alexa Fluor^®^ 488-conjugated goat antibody against rabbit IgG (Molecular Probes, Sigma, St. Louis, MO, USA) at a 1:200 dilution were used as the primary and secondary antibodies, respectively. Immunofluorescence images were collected using a Zeiss LSM 510 META confocal microscope (Analytik Jena AG, Jena, Germany). The samples were excited at 488 nm with a krypton–argon laser, and the emission from the Alexa 488^®^ fluorochrome was detected using a bandpass filter ranging from 505–530 nm.

### 2.5. Phenotype Analysis

Six-day-old seedlings grown in MS medium with or without ABA treatment were positioned upside down after UV-B radiation and grown for 6 days. The phenotypes of treated seedlings were photographed, and the images of their root tips were enlarged using a Zeiss LSM 510 META confocal microscope at the indicated time. UV-B radiation caused leaf chlorosis and root retardation. The phenotypes were analyzed to investigate whether the swollen root tips were regulated by the cortical microtubule orientation/dynamics. Each experimental group contained at least 20 seedlings, and each experiment was repeated at least three times.

### 2.6. The Morphology of Root Cells Staining by PI and Observations

Healthy six-day-old seedlings of *A. thaliana* Col-0 and mutants *sad2-2* and *aba3* were arranged in MS medium containing 0.5 μm L^−1^ ABA for 2 days or imposed by UV-B radiation for 30 min following 1-day normal illumination. Then the root tips were stained with 5 μg mL^−1^ propidium iodide (PI) according to the kit instructions and photographed for morphology observations of the root epidermal cells. The morphology of root cells was visualized to investigate the effects of UV-B radiation on root cells. Each experimental group contained at least 20 seedlings, and each experiment was repeated at least three times.

### 2.7. Statistical Analysis

All values are expressed as the mean ± standard error (SE) (*n* ≥ 10) for each trait. The crooked root lengths and the diameters of the root transition zone of seedlings were measured by the software Image J. All variations between treatments were evaluated with the least significant difference (LSD) multiple range tests (*p* ≤ 0.05) using IBM SPSS Statistics 27 software.

## 3. Results

### 3.1. ABA Strengthens the Adaptive Response to UV-B Stress in A. thaliana

To gain insights into the function of ABA in plants’ response to UV-B stress, the effects of endogenous and exogenous ABA on *A. thaliana* were investigated. Plants of Col-0 and *sad2-2* were used for the study. Col-0 seedlings irradiated with UV-B (90 mJ cm^−2^ for 30 min, which is determined by the pre-experiment (Figure A1) without ABA feeding, displayed remarkable chlorosis and root growth inhibition and subsequently died at 6 DPR (days post-radiation). In contrast, those UV-B-irradiated plants treated with exogenous ABA remained green and mostly survived at 6 DPR (Figure 1a). Moreover, the *sad2-2* mutant survived longer with less chlorosis phenotypic characteristics than Col-0 seedlings under UV-B-irradiation, similar to the ABA-treated Col-0 seedlings (Figure 1b). We, therefore, evaluated the endogenous ABA levels of Col-0 and *sad2-2* under normal and UV-B-radiated conditions. UV-B induced ABA accumulation in both of the plants. Moreover, we noticed that the ABA concentrations of *sad2-2* plants were 19.2 and 46.08 ng g^−1^ FW before and after UV-B radiation, respectively, which were much higher than that of the wild type (5.76 and 22.08 ng g^−1^ FW, respectively) as shown in Figure 1c. Because the *sad2-2* plants were more tolerant to UV-B than the Col-0 (Figure 1a,b), we proposed that the ABA is positively involved in the plant’s response to UV-B radiation.

### 3.2. UV-B Intervenes Root Growth through ABA Signaling

To further uncover if ABA accumulation strengthens the tolerance of plants to UV-B radiation, we used *sad2-2* (high levels of endogenous ABA shown in Figure 1c) and *aba3* (deficient in ABA synthesis) mutants. We analyzed their response to UV-B radiation at the early stages. After exposure to UV-B radiation for 30 min, all the roots of the irradiated seedlings immediately stopped elongating (Figure 2a,b), and at 48 h, the cells of the lower parts of the elongation zone became visibly swollen (Figure 2a). Moreover, compared with the weak response of *sad2-2* and Col-0, *aba3* was hypersensitive to UV-B. The root tips of the *aba3* seedlings expanded into a nodule at 48hr DPR, which was larger than that of *sad2-2* and Col-0. Hence, ABA-deficient mutants *aba3* are not only generally stunted but also more sensitive than *sad2-2* and Col-0 in response to UV-B stress. PI staining data showed that all root tissue of UV-B radiated seedlings at 1 DPR was seriously damaged, and the root epidermal cells of *the aba3* mutant were significantly enlarged (Figure 2c). These findings suggested that ABA deficiency aggravated the growth retardation imposed by UV-B radiation and influenced root morphology by inhibiting elongation and promoting the isotropic expansion of root tips.

### 3.3. UV-B Remodels Microtubule Arrangement in Root Tip

Microtubule dynamics are essential for root tip development. Therefore cortical ar- rays in the elongation zone of root tips were observed at different time points after UV-B stress. As shown in Figure 3, the intact microtubule network of the cell was severely disturbed in Col-0 after UV-B irradiation. Transversely oriented microtubules were completely disordered at 1 HPR, were transformed into short, curved punctate structures, and microtubule abundance decreased sharply. At 2HPR, the microtubule abundance began to increase, and microtubule arrays reappeared obliquely. At 6 HPR, a highly ordered cortical microtubule rearrangement was found to be arranged longitudinally parallel to the longitude axis.

### 3.4. ABA Stabilizes the Microtubule Array and Reduces UV-B-Induced Reorganization

Given that UV-B irradiation resulted in the modulation of both ABA and cortical microtubules in the root, we next examined whether or not there is a potential link between ABA level and microtubule dynamics. The microtubule arrays of the transition zone of the roots were then monitored in the *aba3* and *sad2-2* mutants with or without UV-B radiation. Under normal conditions, the microtubule arrays in the *aba3* mutant were few and scattered compared to Col-0, whereas in *sad2-2,* they were thicker than the wild type. Under UV-B radiation conditions, the cortical array in *aba3* displayed an oblique and lengthwise arrangement which was more severe than the arrangements in Col-0, while in contrast, the influence of UV-B on the cortical array in *sad2-2* was relatively slight expressing as that no lengthwise microtubule was observed (Figure 4). The results revealed that together with ABA, UV-B remodels microtubule dynamics of the root tip, which is supported by the fact that high endogenous ABA levels can stabilize the microtubules and reduce their UV-B-radiation-induced reorganization.

We also evaluated the effect of exogenous ABA on the cortical arrays in Col-0, *aba3,* and *sad2-2*. As shown in the bottom row of Figure 4, the fluorescence intensity in the three genotypes was higher than in untreated controls. More transverse cortical arrays were observed in *aba3* after applying exogenous ABA. Meanwhile, ABA-treated *sad2-2* exhibited relatively thicker and denser, even bunched microtubules, as shown by the arrow, than others due to the higher endogenous ABA content. These results suggested that increased levels of ABA promote the stabilization of transverse cortical microtubules and reduces UV-B stress.

### 3.5. ABA Regulates Primary Root Tip Growth via Stabilizing Cortical Microtubule Arrays

To further explore whether root growth is regulated by ABA and microtubule skeleton, we analyzed primary root morphological characteristics and cortical microtubules of wild-type seedlings after exogenous ABA, taxol, and oryzalin feeding. Taxol is a microtubule stabilizing agent, while oryzalin is a plant dinitroaniline herbicide and a specific depolymerization agent for plant microtubule skeletons. Both ABA- and taxol-treated plants displayed thicker and denser cortical arrays and longer and more root hairs than the control plants. The cortical arrays of 0.25 μmol L^−1^ ABA- and taxol-treated plants were quite similar (Figure 5). In contrast, the primary root of the oryzalin-treated seedlings exhibited a disordered cortical microtubules, resulting in growth inhibition, abnormal root hairs, and apical expansion similar to that of UV-B irradiated seedlings (Figure 2a and Figure 5). These data proved that a low level of ABA promoted primary root elongation and root hair growth by stabilizing the transverse cortical microtubules.

### 3.6. ABA Reduces UV-B-Induced Root Tip Expansion

To better understand the root morphology response to ABA and UV-B radiation, we compared the diameters of the root transition zone of seedlings exposed to taxol, oryzalin, ABA, and UV-B. Statistical analysis revealed that the root elongation inhibition (Figure 2b) and apical expansion caused by UV-B was significantly weakened by ABA application, while there was no significant difference between ABA concentrations of 0.25 μmol L^−1^ and 0.5 μmol L^−1^ (Figure 6). Compared with the treatment of UV-B and oryzalin, low concentrations of ABA can reduce UV-B-induced root tip swelling, similar to taxol.

The results above indicated that both ABA and cortical microtubules are involved in the response to UV-B stress in roots. ABA stabilizes the transverse cortical microtubules and prevents UV-B-induced microtubule reorganization and consequent root tip swelling, which is beneficial for plants in enhancing resistance to UV-B stress.

## 4. Discussion

*A. thaliana’s* roots have become an ideal model for research on plant architecture and developmental plasticity due to its simple structure and sensitivity to environmental stimuli [15,28]. As an underground organ of plants, roots have been ignored or even questioned about the scientific nature of their participation in UV-B response to above-ground light stress. Although generally unexposed to direct sunlight, plant roots also retain systematic signaling mechanisms in response to UV-B. Root *UV-B* sensitive 1 and 2 (*RUS1* and *RUS2*) and four other *RUS* genes have been identified as contributing to the root UV-B signaling [9,31]. Seedling development in *rus* mutants is impeded under UV-B [9]. UV-B is perceived by the UVR8 protein in plants. The expression of UVR8 is also found in roots along with stems and leaves [10]. MYB transcription factors MYB73/77 regulate lateral root growth by interacting with UVR8 in roots in the presence of UV-B by hampering auxin signaling and hence combining UV-B and auxin pathways [12]. In addition, accumulating evidence indicates that root growth is influenced by light through the light-induced release of signaling molecules which can travel from the shoot to the root under natural conditions [28,32,33,34]. Light was efficiently conducted through the stems to the roots in which ELONGATED HYPOCOTYL 5 (HY5) plays an important role. Photo-activated phytochrome B (phyB) can trigger the expression of HY5, a transcription factor, which promotes root growth in response to the expression of HY5 to promote photomorphogenesis [33,34]. Meanwhile, many other pathways are also involved in mediating light-regulated root and shoot growth and development, such as COP1-mediated light signaling, MYB73/MYB77-mediated UV-B signaling, and so on [35]. Thus studies on roots’ response to UV-B stress at the morphological and physiological, cellular, and molecular levels have attracted increasing attention nowadays. Thus that is necessary to question the detailed cytological processes and mechanisms beneath root response to UV-B stress.

As a universal stress hormone, cellular ABA has been shown to mediate plants’ development [15,16,17,30,36,37,38]. ABA is synthesized and accumulated in roots and regulates primary root growth in *A. thaliana* in a concentration-dependent manner, which exogenously applied in low concentrations stimulates, while at high concentrations inhibits root growth [16,17]. ABA plays a vital role in equipping plants for environmental stresses. So, there is a probability of UV-B and ABA signaling crosstalk to manage the combinatorial effect of multiple environmental stresses, although the exact molecular mechanism is yet understudied. ABA biosynthesis mutants showed raised sensitivity toward UV-B stress, which reveals that ABA might be required for UV-B stress resistance. UV-B induces ABA accumulation, which increases the cytosolic Ca^2+^ levels and activates nitric oxide signaling through nitric oxide synthase. The elevated nitric oxide signaling produces an enhanced antioxidant defense response [39]. An ABA-sensitive mutant *sad2* (sensitive to ABA and drought) shows increased UV-B tolerance due to the absence of MYB4 protein in the nucleus and results in the accumulation of UV-absorbing pigments that shield the plant from UV-B radiation [20]. The present study analyzed the effects of endogenous and exogenous ABA in *A. thaliana’s* response to UV-B stress. Results indicate that both endogenous accumulation and exogenous application of ABA can strengthen plants’ adaptive response to UV-B stress (Figure 1, Figure 2 and Figure 6), which agrees with previous reports [5,20]. Interestingly, the possible role of the *SAD2* gene functions not only as an importin mediating MYB4 nuclear trafficking [20] but also in the accumulation of ABA in response to UV-B light, which is supported by the fact that *sad2-2* mutants exhibit excessive ABA accumulation on UV-B exposure (Figure 1c). However, how ABA strengthens plant tolerance to UV-B has yet to be characterized, and research into the role of the ABA signaling pathway in plant responses to UV-B is needed.

The microtubule cytoskeleton is widely involved in many key processes in plant morphogenesis [28,40] and the response to adverse environments [21,22,23,30] for the dynamic features, hence microtubule cytoskeletons are believed to perceive signal transduction in plant stress response [30,41,42,43]. However, the crosstalk and underlying mechanism between microtubules and these signals are still insufficiently studied. In this study, we saw that UV-B radiation caused root tip ectopic outgrowths (Figure 2) and alterations of cortical microtubules (Figure 3). And the microtubule alterations induced by UV-B tended to be attenuated by endogenous ABA. On the contrary, the alterations are sharpened in the ABA-deficient *aba3* mutants with severe root tip expansion (Figure 4). Furthermore, similar to taxol treatment, exogenous application of low-concentration ABA significantly promoted the transverse cortical arrays (Figure 5) to regulate root growth.

Cortical microtubule arrays are essential for determining the growth axis of the elongating cells in plants. Well-ordered transverse microtubule arrays promote cell elongation and restrict cell expansion, with some *A. thaliana* mutants harboring growth defects with unusual cortical arrays accordingly [44,45]. Therefore our results were consistent with previous reports that plants could harden, and ABA treatment increased the stability of tubulin microtubules and reduced the depolymerized action of oryzalin in winter wheat cultivars [46]. ABA signaling also partially rescues the oryzalin-induced (a microtubule-specific destabilizer) alternative root hydrotropism of wild-type *A. thaliana* [30].

While there are also some different results reported. Da Silva et al. [47] found that ABA treatment lowered the abundance of microtubules, altered their orientation, and inhibited embryo cell growth during coffee seed germination. Takatani et al. [42] found that 1 μmol L^−1^ ABA treatment induces ectopic outgrowths in epidermal cells of hypocotyls via promoting cortical microtubule disorganization. Furthermore, Jacques et al. [48] reported that UV-B radiation does not alter the cellular microtubule organization in *A. thaliana* leaves. However, the dynamic rearrangement of microtubules in response to UV-B is spontaneous and fast (Figure 3), which may not be observable over a few days. Besides that, it makes sense that our results differ from those reports due to different processes in different cells instead of root responses to UV-B.

Taken together, we propose that UV-B stress signals trigger endogenous ABA accumulation and cortical microtubule depolymerization, which alter the direction of root growth from elongation to isotropic expansion, as shown in Figure 7. Moderate increased ABA is necessary for protecting transverse cortical arrays to control root growth, which is an adaptive response for *A. thaliana* under ultraviolet-B stress. In addition, ABA accumulation and microtubule reorganization are two antagonistic mechanisms on root morphology, which are simultaneously activated and have significant effects on the plant response to UV-B stress.

It’s worth noting that both UV-B-induced root subapical expansion (Figure 2) and alterations of cortical microtubules (Figure 3) are located in the transition zone, which is important to determine whether root cells will remain in the division zone or transit to elongation and differentiation. And this transition is dependent on precise gradients of phytohormones and the reorganization of microtubules [28]. By questioning the potential molecular mechanisms of the link between ABA and microtubules in *A. thaliana* roots response to UV-B, we further found that *MAP18* (*At5g44610*) expression was regulated by ABA, which is supported by the TAIR database AtGenExpress Visualization Tool (AVT) (Figure A2). The results showed that UV-B stress induced an irreversible inactivation of the quiescent center, which leads to the inhibition of cell proliferation and consequent growth of primary roots. At the same time, MAP18 plays an important role in UV-B-induced lateral root re-generation by regulating quiescent center activity (Figure A3, Figure A4, Figure A5 and Figure A6). All these facts will help us dissect and combine the roles of ABA, UV-B, and microtubule organization in roots in the future.

## Figures and Tables

**Figure 1 genes-14-00892-f001:**
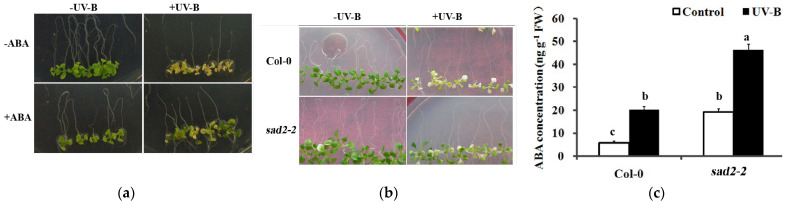
UV-B radiation increased endogenous ABA concentration in *A. thaliana* seedlings, and exogenous ABA supplementation protected seedlings from UV-B radiation. (**a**) Six-day-old wild-type (Col-0) seedlings vertically grown in MS medium with or without 0.5 μmol L^−1^ ABA were irradiated with 90 mJ cm^−2^ of UV-B light for 30 min and positioned upside down. Phenotypes of the seedlings were photographed 6 days post-radiation; (**b**) Six-day-old wild type and *sad2-2* mutant seedlings vertically grown in MS medium were irradiated with 90 mJ cm^−2^ of UV-B light for 30 min and positioned upside down; (**c**) Endogenous ABA levels of six-day-old seedlings from wild type and *sad2-2* mutants were assayed using a Plant Hormone ABA ELISA kit 60 min after radiation began and 30 min after it ended. Each experiment was repeated at least three times. All the statistical analyses were conducted using the IBM SPSS Statistics 27 software. All values are expressed as the mean ± SE (*n* ≥ 20). Letters of a–c represent statistically significant groupings of *p* ≤ 0.05 based on a one-way ANOVA with the LSD multiple range tests.

**Figure 2 genes-14-00892-f002:**
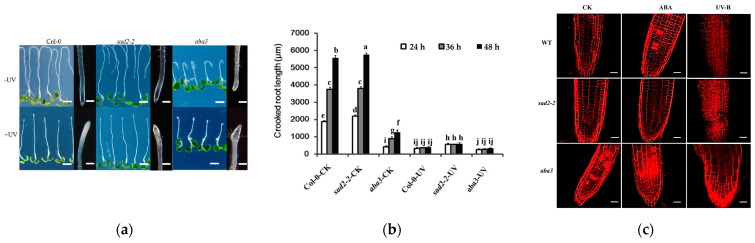
ABA deficiency aggravated the isotropic expansion of root tips induced by UV-B radiation. (**a**) Effects of UV-B radiation on the *A. thaliana* seedling growth. Six-day-old *A. thaliana* wild type (Col-0)*, sad2-2,* and *aba3* seedlings vertically grown in standard MS medium were irradiated with 90 mJ cm^−2^ of UV-B light for 30 min and positioned upside down. The phenotypes of treated seedlings were photographed 48 h post-radiation, and the root tips were enlarged using a Zeiss LSM 510. META confocal microscope. Scale bars represent 2 mm for seedlings and 200 μm for enlarged views of the root tips. (**b**) Effects of UV-B stress on root elongation of wild type and *sad2-2* and *aba3* mutants. The crooked root lengths of seedlings were measured, and those not exposed to UV-B radiation were used as controls (CK). All the statistical analyses were conducted using the IBM SPSS Statistics 27 software. All values are expressed as the mean ± SE (*n* ≥ 20). And letters of a–j represent statistically significant groupings of *p* ≤ 0.05 based on a one-way ANOVA with the LSD multiple range tests. (**c**) The images of the root epidermal cells staining by Propidium Iodide (PI) in three genotype seedlings. Healthy six-day-old seedlings were arranged in MS medium containing 0.5 μmol L^−1^ ABA for two days or imposed by UV-B radiation for 30 min following one day of normal illumination. Then the root tips were stained with 5 μg mL^−1^ PI and photographed to show that ABA deficiency aggravated the root growth retardation and influenced root morphology. Scale bars represent 30 μm.

**Figure 3 genes-14-00892-f003:**
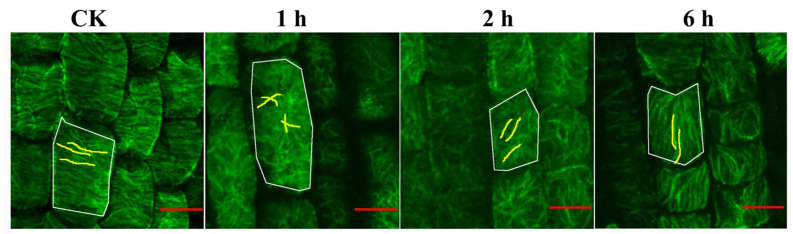
Time series images of the cortical microtubules in root epidermal cells of six-day-old seedlings at 1 h, 2 h, and 6 h post-radiation by UV-B for 30 min in *A. thaliana* wild type (Col-0). Cortical microtubules in root cells of seedlings that did not undergo UV-B radiation were used as controls (CK). The yellow lines processed by software Image J represent the change of length, direction, and structure of the microtubules in the selected cells. Immunofluorescence images were collected using a Zeiss LSM 510 META confocal microscope. Each experimental group had at least 10 seedlings, and each experiment was repeated three times. Scale bars represent 10 μm.

**Figure 4 genes-14-00892-f004:**
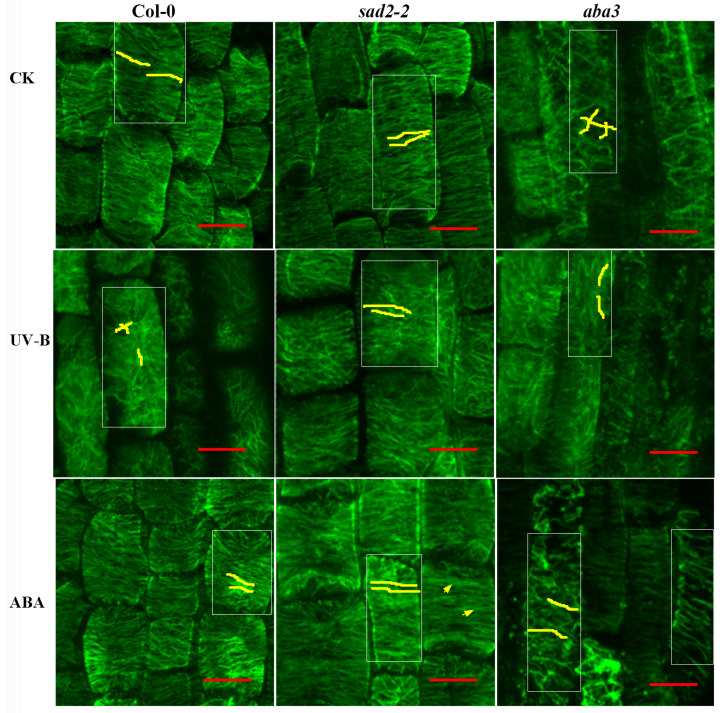
UV-B radiation altered the cortical microtubules in root epidermal cells from seedlings of *A. thaliana* wild type (Col-0), *sad2-2,* and *aba3* mutants, three genotypes. Cortical microtubules were observed 1 h post-radiation by immunofluorescence microscopy in the root epidermal cells of seedlings treated as described above. The yellow lines processed by software Image J represent the change of length, direction, and structure of the microtubules in the selected cells, and the yellow arrows indicate bundles of the microtubules. Immunofluorescence images were collected using a Zeiss LSM 510 META confocal microscope. Each experiment was repeated three times. Scale bars in the fluorescence images of microtubules represent 10 μm.

**Figure 5 genes-14-00892-f005:**
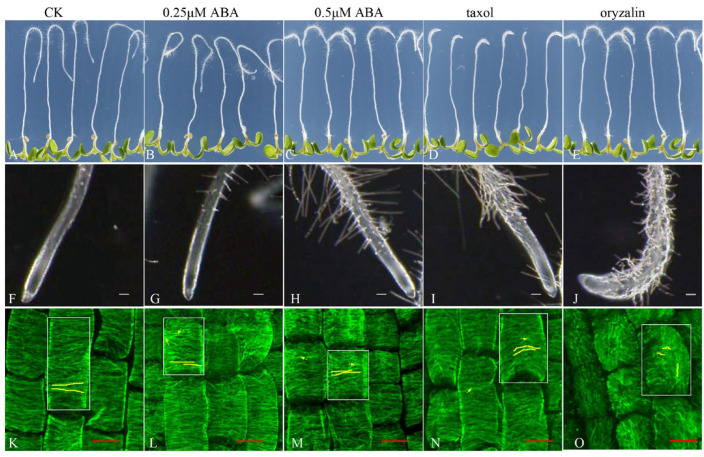
Effects of ABA, taxol, and oryzalin on cortical microtubules of the transition zone of root epidermal cells and phenotypes of *A. thaliana* seedlings. *A. thaliana* grown in an MS medium for 5 days was arranged in an MS medium without drugs as control and in an MS medium containing 0.25 μmol L^−1^ ABA, 0.5 μmol L^−1^ ABA, 0.3 μmol L^−1^ oryzalin or 1 μmol L^−1^ Taxol, respectively, as indicated in the figure. Phenotypes of seedlings (subfigures **A**–**O**), enlarged root tips (subfigures **A**–**O**), and cortical microtubules (subfigures **A**–**O**) were photographed after 2 days, and the images of their root tips were enlarged using a Zeiss LSM 510 META confocal microscope. The yellow lines processed by software Image J represent the change of length, direction, and structure of the microtubules in the selected cells, and the yellow arrows indicate bundles of microtubules. Scale bars represent 1 mm for seedlings, 100 μm for enlarged root tips, and 10 μm for fluorescence views of microtubules.

**Figure 6 genes-14-00892-f006:**
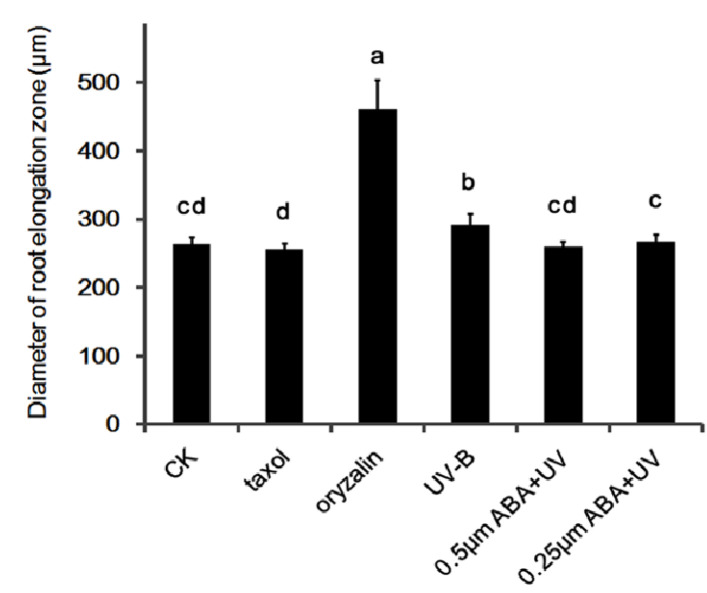
Effects of ABA and UV-B radiation on the root diameter of wild-type seedlings compared to those of taxol and oryzalin treatments. The diameters of the root transition zone of different treated seedlings were measured 24 h post-radiation by image processing software Image J. The data were statistically analyzed by data analysis software IBM SPSS Statistics 27. Untreated seedlings were used as control (CK). Each experiment was repeated three times. All the statistical analyses were conducted using the IBM SPSS Statistics 27 software. All values are expressed as the mean ± SE (*n* ≥ 27). And letters of a–d represent statistically significant groupings of *p* ≤ 0.05 based on a one-way ANOVA with the LSD multiple range tests.

**Figure 7 genes-14-00892-f007:**
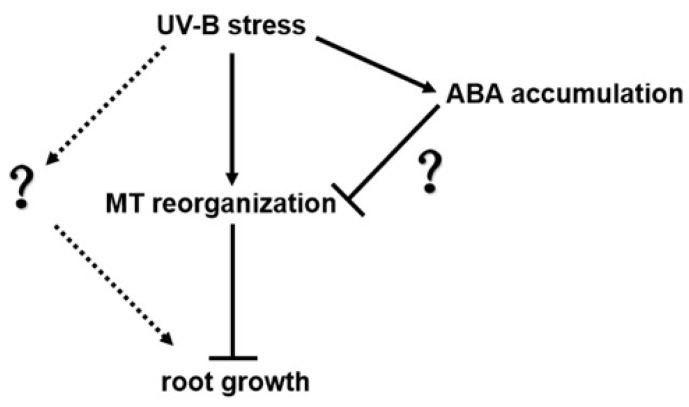
Proposed working model for abscisic acid (ABA) and cortical microtubule (MT)-mediated signaling pathways in *A. thaliana* as a response to UV-B stress. Enhangce 
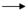
. Repress 
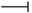
.

## Data Availability

The data supporting the findings of this study are available from the corresponding author upon reasonable request.

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
