# Peer review of "Abscisic Acid Inhibits Cortical Microtubules Reorganization and Enhances Ultraviolet-B Tolerance in Arabidopsis thaliana"

_genes, 2023, doi:10.3390/genes14040892_

Round 1

Reviewer 1 Report

Dear authors,

The manuscript is well written and covers an important area of plant adaptive response to UV-B and presents an additional dimension of ABA- mediated corticle microtubule reorganization under UV-B stress.

For further improvement of the manuscript, I would suggest for a thorough revision with spell check is required as several typos have been detected in most of the sections. Some sentences are incomplete. Avoid using informal language such as "what's more, we care etc" in the text.

The concluding paragraph of the discussion can be strengthened. Expand the abbreviation QC mentioned in the discussion.

Reviewer 2 Report

-All my comments are available in the attached file using the text track changes

-The authors did not use appendix A in the text in a clear way, as they mentioned figures for All figures and Appendices

Reviewer 3 Report

Please use new references.

Reviewer 4 Report

Dear Authors,

Corrections that need to be made were highlighted on the text and notes were added.

Kind regards.

Reviewer 5 Report

The manuscript entitle 'Abscisic Acid Inhibits Cortical Microtubule Reorganization and Enhances Ultraviolet-B Tolerance in Arabidopsis thaliana' by Shi et al. describes the role of ABA in mediating plant response and adaptation under UV-B radiation. 

The manuscript describes an interesting idea, however, clarifications of the following points are required.

1. In general, ABA plays important role in regulating plant response following exposure to stress. However, plant roots do not get direct exposure to UV-B light as compared with the aerial part. The authors must clarify this point. 

2. Propidium iodide stained images need improvement.

3. The authors did not clearly propose any concept or hypothesis. This point should be addressed.

4. The authors must explain how the interplay between ABA and UV-B light might be involved in regulating root growth response under natural condition. 

5. It would be helpful to include one summary figure highlighting the major findings of the study.

6. Careful revision of the English language used is required. 

Reviewer 6 Report

Undoubtedly, the subject is novel, but the presentation of its results and its conclusions must be substantially improved before being published. Particular care must be taken with the images of confocal microscopy photographs on which his main conclusions rest. For me the observations described on these images are not clear enough. Attention should also be paid to some aspects of writing, spaces, capital punctuation, to mention a few. It is notable that the results shown on the map18 mutant and the soverexpressor and RNAi lines for this gene are all in the appendix, as their analysis is good support for their conclusions. Undoubtedly including in your study other mutants in microtubule-associated proteins such as map25, map65 or map200, to mention a few, would help many to improve your observations/conclusions.

Round 2

Reviewer 5 Report

The authors have revised the manuscript appropriately. I have no other query.

Author Response

Thank you very much for the positive comments on our revised manuscript.

Reviewer 6 Report

I really appreciate the attention of the authors in addressing my suggestions and those of other reviewers, as I deduce from the revised version (v2). However, for me the main lack of the article has not been satisfactorily resolved. It is not a problem of resolution of the confocal microscopy images, which are certainly now clearer, if not of interpretation. To facilitate this, the microtubules should be marked, perhaps with an arrow, or even include a diagram of the "normal" pattern so that the images can be properly interpreted. Otherwise it is very difficult to understand what is described and intended to highlight. Additionally, the statistical significances are still missing in Figures 1c and 2b. Another thing that needs to be done is to refer to the appendices in the text. Finally, something that I find interesting to include in the discussion to enrich the report is the possible role of the SAD2 gene (an importin) in the accumulation of ABA in response to UV-B light, more than anything because, as can be seen in the Figure 2C, wild type plants are not affected in accumulating ABA in response to treatment with UV-B light, in fact they accumulate more (3.8x) than the sad2 mutant (2.4x), indicating that although the loss of function of this gene causes super sensitivity to ABA, it does not affect its accumulation in responses to UV-B light.
